# A Comparison of the Level of Acceptance and Hesitancy towards the Influenza Vaccine and the Forthcoming COVID-19 Vaccine in the Medical Community

**DOI:** 10.3390/vaccines9050475

**Published:** 2021-05-08

**Authors:** Magdalena Grochowska, Aleksandra Ratajczak, Gabriela Zdunek, Aleksander Adamiec, Paweł Waszkiewicz, Wojciech Feleszko

**Affiliations:** 1Department of Pediatric Pneumonology and Allergy, Medical University of Warsaw, 02-091 Warsaw, Poland; magdalena.grochowska01@gmail.com (M.G.); olarurarz@gmail.com (A.R.); zdugabriela@gmail.com (G.Z.); aleksander.adamiec@wum.edu.pl (A.A.); 2Doctoral School, Medical University of Warsaw, 02-091 Warsaw, Poland; 3Department of Criminalistics, Faculty of Law and Administration, University of Warsaw, 02-091 Warsaw, Poland; p.waszkiewicz@uw.edu.pl

**Keywords:** COVID-19, vaccine, vaccine hesitancy, healthcare workers, flu vaccine, influenza, SARS-CoV-2

## Abstract

Despite research conducted worldwide, there is no treatment specifically targeting SARS-CoV-2 infection with efficacy proven by randomized controlled trials. A chance for a breakthrough is vaccinating most of the global population. Public opinion surveys on vaccine hesitancy prompted our team to investigate Polish healthcare workers’ (HCWs) attitudes towards the SARS-CoV-2 and influenza vaccinations. In-person and online surveys of HCWs: doctors, nurses, medical students, and other allied health professionals (*n* = 419) were conducted between 14 September 2020 and 5 November 2020. In our study, 68.7% of respondents would like to be vaccinated against COVID-19. The safety and efficacy of COVID-19 vaccinations would persuade 86.3% of hesitant and those who would refuse to be vaccinated. 3.1% of all respondents claimed that no argument would convince them to get vaccinated. 61.6% of respondents declared a willingness to receive an influenza vaccination, of which 83.3% were also inclined to receive COVID-19 vaccinations. Although most respondents—62.5% (262/419) indicated they trusted in the influenza vaccine more, more respondents intended to get vaccinated against COVID-19 in the 2020/2021 season. The study is limited by its nonrandom sample of HCWs but provides a preliminary description of attitudes towards SARS-CoV-2 vaccination.

## 1. Introduction

The severe acute respiratory syndrome coronavirus 2 (SARS-CoV-2) caused a global pandemic with a disease called coronavirus disease 2019 (COVID-19) [1]. The manifestations of COVID-19 range from asymptomatic or mild symptoms to the severe course of the disease leading to death [2]. Several groups are at a greater risk of complications from COVID-19 [3,4,5]. A recent meta-analysis has shown that nearly 10% of COVID-19-positive patients are healthcare workers (HCWs) [6]. A chance for a breakthrough in the fight against the serious consequences of the new disease can be provided by the worldwide vaccination campaign against the SARS-CoV-2 [7]. Despite the scientific community’s unprecedented effort that in 12 months from identifying the new virus developed safe and effective vaccines, a decline in public confidence in vaccines may affect the scale of vaccination and the effectiveness of such prophylaxis [8].

Influenza is another acute respiratory illness. The most effective method for preventing and controlling is a vaccination available for many years [9,10]. During the first months of the COVID-19 pandemic, it was often compared to either seasonal flu or the deadliest flu outbreaks in history due to some mortality and morbidity similarities [11]. It is estimated that globally each year, an average of 389,000 respiratory deaths are associated with influenza (the uncertainty ranges from 294,000 to 518,000) [12]. Despite the yearly death toll and the availability of effective and safe vaccines against influenza, most countries’ vaccination levels seem to be pretty low. Even in the most developed countries, among the most vulnerable age groups (65+), they range from as low as 7.2% (Turkey) to as high as 85.1% (South Korea) [13]. HCWs are no exception in that regard. In Europe, fewer than 30% HCWs vaccinate against seasonal flu [14].

COVID-19 and influenza HCWs vaccination share several similarities. HCWs are facing a higher risk of exposure to viruses responsible for those illnesses than the general population. At different levels, it creates a potential threat to public health. First, infected HCWs may become super-spreaders among the most vulnerable groups, such as those already affected by other illnesses, the elderly, their family members. Second, contracting the virus means absence at the healthcare frontline, which is critical during the pandemic. Finally, HCWs are the role models for most of the general population, and their attitudes and personal decisions may be the key to effective vaccination programs [15].

The growing global vaccine hesitancy phenomenon, similarities between COVID-19 and influenza, and the HCWs significance for the success of any vaccination program prompted our team to investigate the Polish medical community’s attitude towards the upcoming (at the time of the survey) SARS-CoV-2 vaccination and the influenza vaccinations. Informed decision-making in vaccination campaigns is an essential element to solve the current public health crisis.

## 2. Materials and Methods

### 2.1. Sample and Settings

Our study participants were medical doctors (MDs), nurses, physiotherapists, dieticians, and medical students (MS) working or studying in Poland. The inclusion criteria were being a medical doctor, nurse, physiotherapist, dietician, pharmacist, or student attending medical universities. All participants were over 18 years old.

### 2.2. Procedures

The study was performed between 14 September 2020, and 5 November 2020, in two phases. Initially, the questionnaires were administrated in the Pediatric Hospital of the Medical University of Warsaw and the Dermatological Hospital of the Medical University of Warsaw. The respondents completed questionnaires provided on a tablet in the presence of the interviewer. Before completing the questionnaire, each participant was briefly introduced to the topic of the study. A note presenting the main author, subject, and addressees of the survey was on the title page. The process was carried out under a rigorous sanitary regime. Stationary questionnaires constituted 23% of all questionnaires (96/419). The second phase of the study was carried out through an online opt-out survey among medical professionals and medical student groups on Facebook. The online form was introduced due to the lockdown and epidemiological recommendations to construct a representative sample of the surveyed population. Informed consent was obtained from all respondents involved in the study. The questionnaire was conducted according to the Declaration of Helsinki of 1975 and was anonymous.

### 2.3. Data Collection

The questionnaire was based on our previous survey carried out in the general population [8]. Initially, the clarity and comprehensibility of questions and answers were verified by the pilot study on a group of MS. All questionnaires were completed electronically, using Google Forms, by the respondents, either in person on a provided tablet or remotely, via a link opened on the respondent’s devices.

The questionnaire consisted of 12 closed questions about: (i) the effectiveness and (ii) safety of mandatory vaccinations; (iii) attitude towards mandatory vaccinations; (iv) knowledge of being infected with SARS-CoV-2, or (v) knowing someone who is or was infected with SARS-CoV-2; (vi) attitude towards the upcoming SARS-CoV-2 vaccinations and (vii) possible arguments that would persuade the respondent to change their decision; (viii) reimbursement of COVID-19 vaccination; (ix) the amount of money that the respondents would be willing to spend on COVID-19 vaccination; (x) being vaccinated against influenza in the previous season 2019/2020 and (xi) the willingness to receive the influenza vaccination in the current season 2020/2021; as well as (xii) comparison of trust in influenza vaccination and COVID-19. Completing the questionnaire took 5 min on average.

### 2.4. Analyses

Data were collected and processed using Microsoft Excel 2019. Statistical analysis was performed using StatSoft Statistica 13.1.

## 3. Results

### 3.1. Study Group

Among the respondents 79% (331/419) were female, 21% (88/419) were male. The acceptance of the forthcoming COVID-19 vaccine among women and men was 66.8% and 76.1%, respectively. The age of the participants was between 19 and 78. The average age was 27.47 (median 24 years old). The respondents were divided into age groups, the vast majority, of which 60.4% (253/419) were 19–25 years old. The remaining participants were included in the following groups 26–30 years old—22.9% (96/419) of the respondents; 31–40 years old—8.1% (34/419) of the respondents; 41–50 years old—4.8% (20/419) of the respondents; and over 50 years old—3.8% (16/419) of the respondents. MS and HCWs from various professional groups took part in our study. Students constituted the largest part of the sample—57% (239/419). Another material part of the sample was MDs—37.2% (156/419). Nurses—4.3% (18/419) and other allied health professionals (AHP)—1.4% (6/419) were underrepresented in the surveyed sample (Table 1).

### 3.2. Evidence-Based Data on the Safety and Efficacy of COVID-19 Vaccination Is the Primary Convincing Argument to Get Vaccinated among HCWs and MS

A total of 68.7% (288/419) participants of the study answered yes to the question, “Do you intend to receive a COVID-19 vaccination if an effective and safe vaccine is developed?” 21% (88/419) of the respondents were hesitant, while 10.3% (43/419) said they would not get the vaccine. The respondents who answered “no” or “I am not sure” received the follow-up question of what arguments would persuade them to get vaccinated. Most of them, 86.3% (113/131), answered that the results of scientific research confirming the safety and efficacy of vaccination against COVID-19 would persuade them; further arguments that the respondents indicated were “an opinion of an expert, specialist, scientist”—34.4% (45/131) and “possible travel restrictions to those without a confirmed vaccination”—22.9% (30/131). Other arguments included: recommendation of vaccination by the Ministry of Health and Main Sanitary Inspectorate for healthcare workers; recommendation of vaccination by a family doctor; a situation in which a family member or loved one would get vaccinated; if a public figure, from social media, would get vaccinated; low-cost of the vaccine; and a fine for those not vaccinated. The last option: “no argument would convince me to get vaccinated,” was chosen by 13 respondents, representing 3.1% (13/419) of all respondents. Detailed data concerning the above questions and responses are shown in Figure 1.

### 3.3. Acceptance of Flu Vaccination among HCWs and MS Is a Strong Predictor for Attitude towards the Current COVID-19 Vaccination

61.6% (258/419) respondents wanted to receive an influenza vaccination in the 2020/21 season, of which 83.3% (215/258) had previously declared to get COVID-19 vaccination when available. Twenty-one percent (90/419) of the respondents answered that they did not yet know whether they would get a flu vaccination, and 19.6% (71/419) stated that they did not intend to get a vaccination. Declared interest in the future (2020/21) flu vaccination—61.6% is almost twice as big as the declared vaccination rate in the last season (2019/20)—32.9%. Detailed data on COVID-19 and flu vaccination attitudes in the 2020/2021 and 2019/2020 seasons are presented in Table 2.

### 3.4. Lower Age Is Associated with Higher COVID-19 Vaccine Acceptance

The acceptance rate among young respondents was relatively high: 70.8% for those aged 19–25 and 72.9% for those aged 26–30. At the same time respondents aged 41–50 and >50, the willingness to receive the vaccine was 50% (10/20) and 37.5% (6/16), respectively. Regarding the influenza vaccination for the 2020/2021 season, the situation was slightly different. In the youngest group of 19–25 years old, only 56.1% (142/253) presented a willingness to be vaccinated. In the following age groups: 26–30 and 31–40 year-olds, 75% (72/96) and 70% (24/34), respectively, were willing to be vaccinated, while among respondents aged 41–50 and over fifty, 65% (13/20) and 43.8% (7/16), respectively.

### 3.5. A Discrepancy in Vaccination Willingness between Physicians and Nurses

The level of trust in COVID-19 and influenza vaccinations was also compared between various respondents’ professional groups. Among physicians, 73.1% (114/156) were willing to get vaccinated against COVID-19, and 76.9% (120/156) were willing to get vaccinated against influenza in the 2020/2021 season. Among nurses, only 22.2% (4/18) and 33.3% (6/18) were willing to get vaccinated against COVID-19 and influenza, respectively. Among MS, 70.7% (169/239) indicated they would get vaccinated against COVID-19 compared with 54% (129/239), who indicated they would get vaccinated against influenza. The remaining professional groups of the study were too small to be included in the analysis. More detailed information about attitudes towards COVID-19 and flu vaccinations is presented in Table 3.

The respondents were also asked to compare their trust in COVID-19 and influenza vaccinations directly. Most respondents—62.5% (262/419) indicated they trusted in influenza vaccine more, while 26.3% (110/419) indicated they trusted both vaccines equally, while only 3.6% (15/419) trusted COVID-19 vaccination more.

### 3.6. Almost 50% of Respondents Not Vaccinated against the Flu during the 2019/2020 Season Declared an Intent to Get the Flu Vaccine in the 2020/2021

Almost thirty-three percent (138/419) of the respondents reported that they had been vaccinated against influenza in the previous season 2019/2020. The majority of them (93.5%; 129/138) declared to receive an influenza vaccine in 2020/2021, just like 45.9% (129/281) of the respondents, who did not get vaccinated 2019/2020 season. Detailed data on the level of acceptance towards the influenza vaccine in 2019/2021 and 2020/2021 are shown above, in Table 2.

## 4. Discussion

We conducted a study of the potential acceptance of a COVID-19 vaccine in the medical community before such vaccines were available. Among the interviewed, nearly sixty-nine percent responded that they would decide to vaccinate if it were proven safe and effective, and 86.3% of hesitating and refusing respondents said that they would get vaccinated if scientific research confirmed the effectiveness of the vaccine. The second most chosen option was “expert, specialist or scientist opinion”. It is worth noting that scientific data itself, rather than expert opinion, was most likely to convince a hesitant participant, highlighting the lowering trust in expert opinions, as well as a willingness to search for and analyze scientific data among HCWs [16,17]. That reveals reasonable concern to the safety and effectiveness and significant fears associated with new vaccines. Only 3% (13/419) of all respondents declared they did not intend to receive the COVID-19 vaccination. Sixty-two percent of respondents (258/419) intended to get influenza vaccination in the 2020/2021 season.

This hardly unanimous willingness to accept COVID-19 and influenza vaccines is a cause for concern. Observing the trends among the age groups, we could notice a decrease in confidence in the upcoming COVID-19 vaccination with increasing age metrics. Professions in which a high tendency toward acceptance was observed tended to be MS and doctors. A relatively low rate of acceptance has been shown among nurses.

Declared interest in the future (2020/21) flu vaccination—61.6% is almost twice as big as the declared vaccination rate in the last season (2019/20)—32.9%. The reason for such increased influenza vaccine interest may be the COVID-19 death toll. However, it may also be an artifact caused by the observer-expectancy effect or social desirability bias [18]. Studies comparing declarations with real-life vaccination rates may confirm the latter. Antczak and colleagues surveyed HCWs and found that 81% of them declared intent on getting the flu vaccine, and only half of that group declared being vaccinated regularly—38% [19]. When checked in independent and reliable sources, the documented vaccination rate was six times smaller (!), reaching 6% [20]. Therefore, such declarations may be a serious indication of what HCWs think is appropriate or right rather.

In 2019, the World Health Organization named vaccine hesitancy among the top ten threats to global health [21]. Many studies were reflecting the attitudes towards vaccinations in the general population. For example, Lazarus et al., in their recent report on potential acceptance of a COVID-19 vaccine in 19 countries (including China, Russia, UK, US, France, and Poland), collected responses from 13 thousand respondents [22]. Polish respondents reported the highest negative responses (27.3%) and only 56.31% of positive answers. Lower acceptance rates were present only in Russia. Correspondingly, in our recent study, we have seen even more distressing numbers of only 37% of Polish respondents, who showed a willingness to be vaccinated with the forthcoming COVID-19 vaccine [8].

Interestingly, these results differ from the ones obtained in our survey (68.7%), and the one by Szmyd et al. (82.95%) studies among Polish HCWs and show a visible discrepancy between medics and the general population [23]. Moreover, 50% of the respondents in our previous study with negative attitudes to the COVID-19 vaccination rejected all arguments and remained unconvinced. In contrast, among HCWs, these standpoints remain in the minority with around 3–4% of ”definitely no” answers [24]. We agree with Detoc et al. that HCWs are more willing to get vaccinated against COVID-19 than non-HCWs. According to their study, the proportion of HCWs willing to receive a jab was 81.5%, and this proportion in non-HCWs was 73.7%. On the other hand, there are reports from the United States, Canada, and Europe about concerns among medics about the new COVID-19 vaccines [25,26]. More detailed studies could help understand those differences and identify responsible factors.

Our research has shown that occupational status influences vaccine acceptance. Similarly, in the studies of Grech et al. and Dror et al., doctors were more likely to take both the influenza vaccine and the forthcoming COVID-19 vaccine [27,28], an effect due to respective knowledge of the topic in this group compared to other groups of HCWs [29,30,31]. It is worth indicating that the nurses who participated in our study reached the level of acceptance for COVID-19 and influenza vaccination of 22% and 33%, respectively. However, these results must be interpreted with caution, knowing substantial cultural interference upon these results with, e.g., 63% and 49% acceptance for COVID-19 and influenza among Chinese nurses [32]. The level of acceptance towards the vaccine was high among MS. Vaccine acceptance in this group was over 70%, almost as high as in the group of MDs, which is consistent with other studies [33,34].

Regarding sex distribution of the COVID-19 vaccine acceptance, our study shows a higher willingness of male individuals to receive the vaccination, which correlates with the foregoing studies [8,27,35]. Additionally, the disproportionate group sizes of female—331/419 and male—88/419 respondents could have affected the results.

Another interesting aspect of our study is the impact of the COVID-19 pandemic on the acceptance of seasonal influenza vaccination. Here we show that the acceptance rate of seasonal influenza increased between 2019 (32.9%) and 2020 (61.6%). The correlation between HCW’s acceptance toward the influenza vaccine and the COVID-19 vaccine was also identified in the study by Kose et al. [36]. However, this estimation is different from the other surveys involving HCWs’ populations. According to the studies conducted by Gagneux-Brunon et al. and Grech et al. the respondents, who had the intention to get the flu vaccine during the following season accounted for 54.6% and 69%, respectively, while the vaccine rate during the previous season was 57.3% and 49%, respectively [27,35]. On the other hand, Di Pumpo et al. show results with a marked increase in the respondent’s willingness to receive influenza vaccine between the 2019/20 season (24.19%) and 2020/21 season (54.46%) [37].

On 11 December 2020, the advisory committee on immunization practices (ACIP) recommended, as interim guidance, that healthcare personnel be offered the COVID-19 vaccine in the initial phase of the vaccination program [38]. Regarding these recommendations and national vaccination programs designed for healthcare professionals and MS, the vaccination uptake level proved high. After two months of vaccination, in 2021, the Polish Ministry of Health announced 94% of MDs and 80% of nurses as vaccinated against COVID-19 [39]. Simultaneously, several studies were published, confirming the relatively high safety and efficacy of COVID-19 vaccinations [40,41,42]. Correspondingly, in our survey, 86.3% of respondents admitted that scientific evidence on vaccine safety and efficacy would be the most persuasive. We expect that this unquestionable success will contribute to the increase in acceptance and will minimize vaccination hesitancy in the general population.

This study has several limitations. First of all, it must be considered that the reported surveys are executed at a certain point in time. Survey questions, by nature, are vulnerable to misinterpretation by individual participants, especially those filling in the answers on their own. This particular survey was conducted in the context of an emerging and rapidly evolving situation. Day-to-day variations in perceived disease threat and COVID-19 vaccine development may have influenced the respondents’ answers. Second, given the hypothetical nature, the study results may differ from actual practice, and some self-reported answers may lead to information bias. We asked the respondents to report their intention to receive the influenza vaccine and the COVID-19 vaccine if it is available in the future. A considerable number of study participants (21.0% and 21.5%) reported “Not sure” about their intention to receive the COVID-19 and influenza vaccinations, respectively. The real intention could be different when the vaccine is available. The study’s main limitation was the nonrandom volunteer sample of survey participants, which made it impossible to calculate a participation rate or confidence limits of the observed proportions. In-person answers were collected in university hospitals in Poland’s capital and largest city. At the same time, online questionnaires would only be available to medical workers frequenting social media, which constitutes a limitation, despite our efforts to publish the questionnaire on diverse groups and allowing it to circulate for over a month. For these reasons, the sample only represents a subset of the target population. Any generalizations of the findings to the entire population of HCWs are subject to these caveats.

Moreover, using only an offline-exclusive survey was not feasible during the pandemic period (lockdown due to COVID-19). The online survey may limit the representativeness of the study sample. Finally, the number of nurse participants was low and may not reflect the broader nursing community’s opinions in Poland. It does, however, reveal an unsettling trend of hesitancy towards vaccinations, especially considered together with the results of the studies mentioned above.

Despite its limitations, our study provides insight into the attitudes towards vaccination among HCWs. We consider this study particularly important in the Polish population, which is highly hesitant regarding the COVID-19 vaccination. Moreover, the study undoubtedly has an educational and practical potential for the general population and public health pursuits. It identifies fears associated with vaccination among the group that is critical for the effective vaccination campaign. Addressing them properly by offering scientific evidence supported by opinion leaders and public intellectuals may convince the hesitant group.

Further research would help understand vaccination hesitancy better since it is one of the most current threats to preventing infections, especially at the COVID-19 pandemic.

## 5. Conclusions

The vast majority of HCWs in our study (68.7%) expressed willingness to receive a COVID-19 vaccination, and over 60% declared readiness for influenza vaccination in the next season. These data suggest a higher acceptance of vaccines among Polish HCWs compared to Poland’s general population. However, there are many divergences amid HCWs that should be addressed by public health activity during the next months of the new COVID-19 vaccine’s distribution. Our study identifies those groups of Healthcare Providers most hesitant to get vaccinated, towards whom the bulk of the promotional efforts should be directed. As suggested by our results, promotional materials intended for HCWs should consist of scientific evidence and expert opinions. We expect the results of our study to positively impact vaccination coverage, both in the general population and in the medical community. We believe our results provide a valuable contribution to the debate on the acceptance and hesitation towards COVID-19 and influenza vaccinations.

## Figures and Tables

**Figure 1 vaccines-09-00475-f001:**
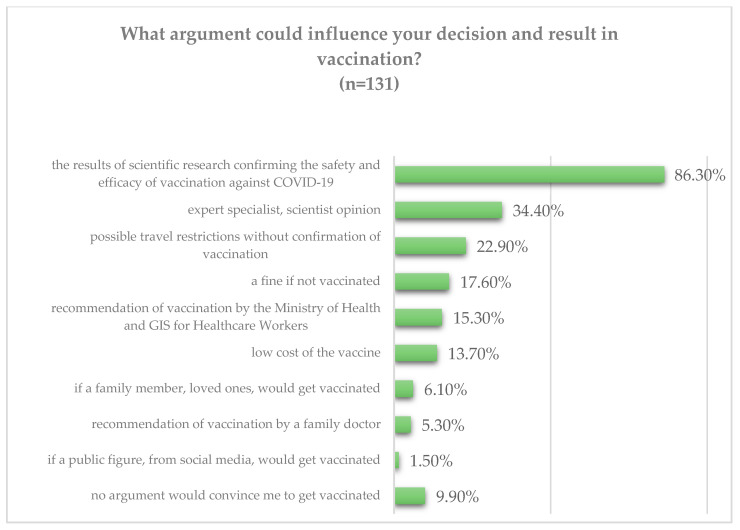
Potential arguments that might convince the hesitant respondents to receive a COVID-19 vaccine (*n* = 131).

**Table 1 vaccines-09-00475-t001:** Study group characteristics (*n* = 419).

Variables	Total	Profession Groups; *n* (%)			
		Medical Students	Medical Doctors	Nurses	Other AHP *
Total; *n* (%)	419 (100)	239 (57.0)	156 (37.2)	18 (4.3)	6 (1.4)
Male; *n* (%)	88 (21)	50 (56.8)	38 (43.2)	0	0
Mean age (range)	27.47 (19–78)	23 (19–31)	34 (24–78)	46 (22–57)	28 (25–32)
Age groups					
19–25	253	220 (87)	28 (11.1)	2 (0.8)	3 (1.2)
26–30	96	17 (17.7)	77 (80.2)	0	2 (2.1)
31–40	34	2 (5.9)	28 (82.4)	3 (8.8)	1 (2.9)
41–50	20	0	13 (65)	7 (35.0)	0
over 50	16	0	10 (62.5)	6 (37.5)	0

* AHP—allied health professionals.

**Table 2 vaccines-09-00475-t002:** A comparison of the level of acceptance and hesitancy towards the COVID-19 vaccine and influenza vaccine.

Variables	Total; *n* (%)	Do You Intend to Get a Flu Vaccination in the 2020/2021 Season?; *n* (%)		
		Yes	Not Sure	No
Total; *n* (%)	419 (100)	258 (61.6)	90 (21.5)	71 (16.9)
Do you intend to get a COVID-19 vaccination if an effective and safe vaccine is developed?				
Yes	288 (68.7)	215 (83.3)	53 (58.9)	20 (28.2)
Not sure	88 (21)	36 (14)	29 (32.2)	23 (32.4)
No	43 (10.3)	7 (2.7)	8 (8.9)	28 (39.4)
Did you get a flu vaccine during the previous 2019/2020 season?				
Yes	138 (32.9)	129 (93.5)	8 (5.8)	1 (0.7)
No	281 (67.1)	129 (45.9)	82 (29.2)	70 (24.9)

**Table 3 vaccines-09-00475-t003:** Attitudes towards COVID-19 and flu vaccination in 2020/2021 season among study groups.

Variables	Total; *n*	Do You Intend to Get a COVID-19 Vaccination If an Effective and Safe Vaccine Is Developed?; %			Do You Intend to Get a Flu Vaccination in the 2020/2021 Season?; %		
		Yes	Not Sure	No	Yes	Not Sure	No
Total; *n* (%)	419 (100)	288 (68.7)	88 (21)	43 (10.3)	258 (61.6)	71 (16.9)	90 (21.5)
Gender							
Male	88	76.1	15.9	8	64.8	23.9	11.4
Female	331	66.8	22.4	10.9	60.7	20.8	18.4
Profession							
Medical students	239	70.7	19.7	9.6	54	28.5	17.6
Medical Doctors	156	73.1	19.2	7.7	76.9	10.3	12.8
Nurses	18	22.2	50	27.8	33.3	33.3	33.3
other AHP *	6	33.3	33.3	33.3	33.3	33.3	33.3
Age groups							
19–25	253	70.8	19	10.3	56.1	25.3	18.6
26–30	96	72.9	17.7	9.4	75	14.6	10.4
31–40	34	67.6	29.4	2.9	70.6	14.7	14.7
41–50	20	50	35	15	65	20	15
over 50	16	37.5	37.5	25	43.8	18.8	37.5

* AHP—allied health professionals.

## Data Availability

Data are available upon request.

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
