# Peer review of "A Comparison of the Level of Acceptance and Hesitancy towards the Influenza Vaccine and the Forthcoming COVID-19 Vaccine in the Medical Community"

_vaccines, 2021, doi:10.3390/vaccines9050475_

Round 1

Reviewer 1 Report

In the manuscript titled “A comparison of the level of acceptance and hesitancy towards the influenza vaccine and the forthcoming COVID-19 vaccine in the medical community” by Magdalena et.al., the authors described an important survey-based statistical study to give an idea about the Polish medical community’s attitude towards the SARS-CoV-2 and influenza vaccinations. The authors did the survey and statistical analysis in a very systematic way to get to the conclusion. The factors and the explanations provided in the manuscript are very logical.

However, this reviewer has a few comments regarding some of the data in the manuscript. Those are as follows.

  1. In the "3.1 Study group" section, the authors mentioned that they had surveyed a much higher percentage of female respondents than male and there was a significant gap (~ 9.7%) in vaccination acceptance between them. Therefore, this author believes that reporting 68.7% vaccination acceptance, in general, can be misleading. Instead, it is suggested to report male and female vaccination acceptance percentages separately.

  1. In the "3.2" section, the authors reported a significant percentage gap between "the results of scientific research confirming the…" and "expert specialist, scientist opinion. The authors did not discuss any plausible reasons behind this. This reviewer thinks that both the parameters are close enough to have nearly the same percentages as both involve opinion and analysis from a scientific society. That is why the reviewer suggests the author to cite a plausible explanation for this reported fact.

  1. In the" Discussion" section line 219, the authors claimed to have "independent and reliable sources" to suggest that the documented vaccination rate was six times smaller than the reported percentage in reference 18. This reviewer believes that this significant reduction in percentages is an important piece of information. That is why this reviewer would like the authors to include a reference for their "independent and reliable source".

Other than the above-mentioned comments, this manuscript is a good fit for acceptance in Vaccine.

Author Response

Response to Reviewer 1 Comments

Point 1: In the "3.1 Study group" section, the authors mentioned that they had surveyed a much higher percentage of female respondents than male and there was a significant gap (~ 9.7%) in vaccination acceptance between them. Therefore, this author believes that reporting 68.7% vaccination acceptance, in general, can be misleading. Instead, it is suggested to report male and female vaccination acceptance percentages separately.

Response 1: We appreciate the reviewer’s point that the disproportion in vaccination acceptance between female and male respondents should be reported more markedly. However, we believe that presenting these results separately could influence the work clarity and make the study difficult to relate to other scientific research papers (where the acceptance level is generally reported as the average percentage of all respondents). Nevertheless, we do agree that this disproportion should be marked more clearly. Therefore, we highlighted this issue in the discussion, paying additional attention to the disproportionate sizes of the groups of female - 331/419 and male - 88/419 respondents, which could have affected the results and we showed the correlation in the distribution of the acceptance between women and men with similar studies in the area of COVID-19 vaccines. (P8; L263-266)

Point 2: In the "3.2" section, the authors reported a significant percentage gap between "the results of scientific research confirming the…" and "expert specialist, scientist opinion. The authors did not discuss any plausible reasons behind this. This reviewer thinks that both the parameters are close enough to have nearly the same percentages as both involve opinion and analysis from a scientific society. That is why the reviewer suggests the author to cite a plausible explanation for this reported fact.

Response 2: We thank the reviewer for his insightful comment. It is true that both those options could be described as “input from the scientific community”. The authors, however, placed a clear distinction within the questionnaire between scientific data and expert opinions. It is an important distinction that we do not feel it is justified to forgo. As suggested by the reviewer, we have added a short segment of discussion on the topic, which might elucidate the possible reasons for the observed differences. (P7; L202-206)

Point 3: In the" Discussion" section line 219, the authors claimed to have "independent and reliable sources" to suggest that the documented vaccination rate was six times smaller than the reported percentage in reference 18. This reviewer believes that this significant reduction in percentages is an important piece of information. That is why this reviewer would like the authors to include a reference for their "independent and reliable source".

Response 3: We thank the reviewer for pointing out this undeniable omission. We attached additional citations to these missing sources and corrected the references for the statements from the whole paragraph. (P7; L225-228)

Thank you again for pointing out all these errors. We have corrected them all.

We wish to thank the reviewer for his thorough commentary and we hope that you will find our manuscript of a better form more suitable for publication.

Yours faithfully,

Wojciech Feleszko MD, PhD

Reviewer 2 Report

The study is limited by the non-random (on-line, volunteer) sampling method. However, the findings could be of use to some readers, therefore, in my opinion the manuscript could be published, if the limitations are clearly stated, and the authors refrain from making over-statements. In particular, be very careful about making statements alleging statistical significance.

P1, L21: Replace "a significant part of" with "most".

P1, L23: Append to the abstract a sentence to the effect of, "The study is limited by the non-random, volunteer sample of HCWs, but it provides preliminary description of attitudes towards SARS-CoV-2 vaccination."

P1, L105: Replace "differences between the surveyed groups" with "associations of vaccination intentions with hypothesized predictors".

P3, L117: Replace "significant" with "material".

Table 2: The p-values do not make sense. If you are testing the null hypothesis that intention to get COVID-19 vaccination is not associated with intention to get flu vaccination in 2020/2021, there should be one p-value, not three. Similarly, if you are testing the null hypothesis that having received flu vaccination in 2019/2020 is not associated with intention to get flu vaccination in 2020/2021, there should be one p-value, not three. Correct these values, or remove the p-values entirely from Table 2, as they do not serve much useful purpose in any case.

P5, L176: Replace "A significant part of" with "Most".

Table 3: As in Table 2, the p-values do not make sense. There should be six p-values, not twenty-two. Correct these values, or remove the p-values entirely from Table 3, as they do not serve much useful purpose in any case.

P7, L198: Replace "85.4%" with "86.3%".

p7, L202-203: The last sentence of the paragraph with , "Sixty-two percent of respondents(258/419) Intended to get influenza vaccination in the 2020/2021 season."

P7, L208-209: Delete the sentence, "The difference was statistically significant for respondents over 50 years of age (p = 0.02)", as it does not follow any ccomprehensible hypothesis test.

P7, L232: Delete the word "significantly".

P8, L257: Delete the word "significantly".

P8, L292-293: Delete the last sentence of the paragraph because it is incorrect. Sample size is irrelevant. The diversity and representativeness of the sample are determined by the sampling strategy, i.e., stratification and randomness, techniques which you did not employ.

P8, L299: Insert a paragraph to the effect of, "The main limitation of the study was the non-random, volunteer sample of survey participants. Because of this, it is not possible to calculate a survey participation rate, it is not possible to calculate confidence limits of the observed proportions, and one does not know what subset of the target population the participants represent. Any generalizations of the findings among the participants to the entire population of HCWs are subject to these caveats."

P8, L300: Replace the phrase "has many strengths" with "provides information about attitudes towards vaccination among HCWs", then delete the rest of the sentence. As I commented above, sample size is irrelevant if the sample is not random.

P9, L313: Replace "70%" with "68.7%".

P9, L315: Replace "clearly illustrate the" with "suggest a"

Author Response

Response to Reviewer 2 Comments

Point 1: P1, L21: Replace "a significant part of" with "most".

Response 1: This phrase has been corrected as suggested by the reviewer. (P1; L21)

Point 2: P1, L23: Append to the abstract a sentence to the effect of, "The study is limited by the non-random, volunteer sample of HCWs, but it provides preliminary description of attitudes towards SARS-CoV-2 vaccination."

Response 2: We thank the reviewer for this valuable comment. Indeed, this limitation is relevant to the interpretation of our results and essential to understanding the message of our study. We have added the suggested sentence and have re-written the abstract to incorporate this sentiment while complying with the 200-word limit (P1; L24-25)

Point 3: P1, L105: Replace "differences between the surveyed groups" with "associations of vaccination intentions with hypothesized predictors".

Response 3: Thank you kindly for your insightful comment. We have followed the suggestion and we modified the sentence as proposed by the reviewer. (P3; L107-108)

Point 4: P3, L117: Replace "significant" with "material".

Response 4: We corrected the phrase as suggested by the reviewer. (P3; L120)

Point 5: Table 2: The p-values do not make sense. If you are testing the null hypothesis that intention to get COVID-19 vaccination is not associated with intention to get flu vaccination in 2020/2021, there should be one p-value, not three. Similarly, if you are testing the null hypothesis that having received flu vaccination in 2019/2020 is not associated with intention to get flu vaccination in 2020/2021, there should be one p-value, not three. Correct these values, or remove the p-values entirely from Table 2, as they do not serve much useful purpose in any case.

Response 5: We thank the reviewer for their valuable input. As the reviewer had suggested, we decided to omit the p-values altogether, seeing as they did not contribute much to the presentation of our results. (P5; Table 2.)

Point 6: P5, L176: Replace "A significant part of" with "Most".

Response 6: We corrected this phrase as suggested by the reviewer. (P5; L179)

Point 7: Table 3: As in Table 2, the p-values do not make sense. There should be six p-values, not twenty-two. Correct these values, or remove the p-values entirely from Table 3, as they do not serve much useful purpose in any case.

Response 7: Once again, we thank the reviewer for their feedback. As the reviewer had suggested, we decided to omit the p-values altogether, seeing as they were not crucial to the presentation of our results. Additionally, we have added a missing value (56.1) to the Table 3. that was accidentally overlooked. (P6; Table 3.)

Point 8: P7, L198: Replace "85.4%" with "86.3%".

Response 8: We appreciate the reviewer’s vigilance and attention to detail. We have corrected the lapse according to the comment. (P6; L201)

Point 9: P7, L202-203: The last sentence of the paragraph with, "Sixty-two percent of respondents (258/419) Intended to get influenza vaccination in the 2020/2021 season."

Response 9: This has been amended as suggested. (P7; L209-210)

Point 10: P7, L208-209: Delete the sentence, "The difference was statistically significant for respondents over 50 years of age (p = 0.02)", as it does not follow any comprehensible hypothesis test.

Response 10: We deleted this sentence as suggested by the reviewer. (P7; L216-217)

Point 11: P7, L232: Delete the word "significantly".

Response 11: We corrected this phrase as suggested by the reviewer. (P7; L240)

Point 12: P8, L257: Delete the word "significantly".

Response 12: This phrase also has been corrected as suggested by the reviewer. (P8; L269)

Point 13: P8, L292-293: Delete the last sentence of the paragraph because it is incorrect. Sample size is irrelevant. The diversity and representativeness of the sample are determined by the sampling strategy, i.e., stratification and randomness, techniques which you did not employ.

Response 13: We thank the reviewer for this comment and agree with his opinion that the size of the group was irrelevant to obtaining the representativeness of the sample. Nevertheless, our team made every effort to reach a varied group of respondents by using social media as a tool to maximize diversity of the Healthcare Workers representatives. Hence, we have deleted the sentence in question and added a few words of explanation in the previous sentence, to illustrate our methods. (P9; L308-310)

Point 14: P8, L299: Insert a paragraph to the effect of, "The main limitation of the study was the non-random, volunteer sample of survey participants. Because of this, it is not possible to calculate a survey participation rate, it is not possible to calculate confidence limits of the observed proportions, and one does not know what subset of the target population the participants represent. Any generalizations of the findings among the participants to the entire population of HCWs are subject to these caveats."

Response 14:  We kindly thank the reviewer for their insight. We have incorporated the suggestions into the limitations paragraph to maintain the coherence of the text. (P9; L303-305, 311-313)

Point 15: P8, L300: Replace the phrase "has many strengths" with "provides information about attitudes towards vaccination among HCWs", then delete the rest of the sentence. As I commented above, sample size is irrelevant if the sample is not random.

Response 15: Thank you for this comment. We have adjusted the sentence to be in accordance with the sentiment of the reviewer. (P9; L320-321)

Point 16: P9, L313: Replace "70%" with "68.7%".

Response 16: We thank for this comment. This lapse has been corrected accordingly. (P9; L334)

Point 17: P9, L315: Replace "clearly illustrate the" with "suggest a".

Response 17: We corrected this phrase as suggested by the reviewer. (P9; L336)

Thank you again for pointing out all these errors. We have corrected them all.

We wish to thank the reviewer for his thorough commentary and we hope that you will find our manuscript of a better form, more suitable for publication.

Yours faithfully,

Wojciech Feleszko MD, PhD

Round 2

Reviewer 2 Report

I advise the following minor edit before publication.

P1, L104-106: Delete the sentence that begins with, "Pearson's chi-squared test" because it is immaterial since you removed the p-values from Tables 2 and 3.
